# Fusing LLMs with Scientific Literature for Heuristic Discovery

## Abstract

The automated design of high-performance algorithms, particularly for NP-hard optimization problems, remains a significant challenge. While Large Language Models (LLMs) demonstrate remarkable code generation capabilities, their reliance on internalized, general-purpose knowledge often limits their efficacy in crafting sophisticated, domain-specific heuristics. This paper introduces Knowledge-Augmented Evolutionary Algorithm Design (KA-EAD), a novel framework that synergistically integrates LLMs with evolutionary computation and dynamic knowledge retrieval from scientific literature. KA-EAD orchestrates a co-evolutionary process where LLMs act as intelligent generative and mutative operators. Crucially, at strategic junctures, the system formulates queries based on intermediate evolutionary artifacts (e.g., LLM-generated reflections) to retrieve pertinent 'knowledge chunks' from a curated, domain-specific corpus. This retrieved, verifiable knowledge is then injected into the LLM's context, guiding it to generate more informed and effective algorithmic solutions. By explicitly grounding the LLM's creative process in external scientific insights, KA-EAD transcends the limitations of relying solely on pre-trained knowledge, enabling a more targeted and robust exploration of the heuristic design space. It showcases a step towards AI systems that can actively learn from and build upon human scientific progress. Code is available at **Supplement**.

## 1 Introduction

The pursuit of efficient algorithms has perennially driven computer science, laying the groundwork for transformative advancements across diverse disciplines (Cormen et al., 2022). Crafting novel solutions, especially for NP-hard optimization problems, has traditionally been a human-centric endeavor reliant on deep expertise and arduous iterative refinement, a paradigm facing inherent scalability limits when exploring vast algorithmic design spaces (Garey & Johnson, 1979; Falkenauer, 1996; Anken & Beasley, 2012; Chakhlevitch & Glass, 2009; Taillard, 1993; Beasley, 1988). The advent of Large Language Models (LLMs) offers unprecedented potential for automating and augmenting this process, given their remarkable ability to translate specifications into code (Chen et al., 2021b; Li et al., 2022b). However, when faced with the *de novo* design of sophisticated high-performance heuristics, LLMs—trained on general corpora—often lack the specialized understanding, nuanced implementation insights (e.g., for tensor-based metaheuristics), and the crucial empirically-grounded iterative loop characteristic of expert algorithm design (Brown et al., 2020).

Pioneering efforts have sought to bridge this gap by integrating LLMs with Evolutionary Computation (EC), demonstrating that LLMs can act as powerful variation operators or facilitate co-evolution of heuristic concepts and code (Romera-Paredes et al., 2024; Liu et al., 2024; Ye et al., 2024). These approaches mark a significant shift towards a collaborative LLM-EC paradigm for heuristic discovery. However, a fundamental limitation of these methods is their primary reliance on the LLM's **internalized** knowledge. While formidable, this knowledge can be static, incomplete, or lack the fine-grained, up-to-date details crucial for cutting-edge heuristic design, especially in specialized or rapidly advancing algorithmic sub-domains. Consequently, guiding LLMs to explore truly novel and high-quality regions of the vast heuristic design space without explicit grounding in external, verifiable knowledge remains a significant impediment, potentially hindering the discovery of groundbreaking or highly optimized solutions. This paper addresses the critical challenge of

*infusing LLM-driven evolutionary algorithm design with explicit, dynamic, and targeted knowledge harvested from domain-specific scientific literature.* We posit that the full potential of LLMs in algorithm design can be realized by enabling them to 'consult' and integrate insights from a curated knowledge base of relevant research throughout the evolutionary process. Our core aim is to transcend the LLM's inherent knowledge boundaries, transforming its creative generation from an often unguided exploration into a more informed, evidence-backed synthesis.

To this end, we introduce a novel framework that orchestrates a *synergistic co-evolutionary process deeply integrated with dynamic knowledge retrieval and augmentation.* As illustrated in Figure 1, our approach centers on an iterative evolutionary loop where LLMs generate, evaluate, and refine candidate algorithmic heuristics. *The pivotal innovation is a dynamic knowledge-augmented generation operator.* This operator, activated at critical junctures, uses intermediate evolutionary artifacts (e.g., LLM-generated reflections or improvement hypotheses) as queries to retrieve the most pertinent 'knowledge chunks' from a pre-curated, problem-relevant scientific literature corpus. These retrieved textual segments are then injected directly into the LLM's prompt, enriching its context beyond immediate evolutionary history or internal knowledge. Thus equipped, the LLM is guided to generate new or substantially modified heuristic code, with meticulous prompt engineering ensuring implementation correctness and computational efficiency, particularly for complex structures like tensor-based operations. This knowledge-infused process iteratively steers the search towards increasingly sophisticated and effective algorithmic solutions.

This work makes three primary contributions:

❶ *A New Paradigm for Heuristic Discovery that Overcomes LLM Knowledge Limitations.* We introduce and validate a novel framework, Knowledge-Augmented Evolutionary Algorithm Design (**KA-EAD**), which directly addresses the fundamental bottleneck of LLM-based algorithm design: its reliance on static, pre-trained knowledge. As foundational work, we establish an automated pipeline for literature screening, knowledge chunking, and vectorized indexing. This provides a robust foundation for dynamically grounding the LLM in an external, verifiable corpus of scientific literature, enabling it to generate heuristics that are not only syntactically correct but also informed by cutting-edge, domain-specific insights, thereby transcending the inherent boundaries of the model's internalized knowledge.

❷ *An Innovative Method for Transforming 'Internal Reflection' into 'Active Knowledge Seeking'.* We transform the reflection mechanism from a closed-loop tool for fine-tuning existing solutions into an *engine that drives the evolutionary process to actively seek breakthrough knowledge from external sources.* Specifically, our method leverages reflection ($\rho_{ST}, \rho_{LT}$) to identify the current population's 'knowledge gaps' and performance bottlenecks. These deep insights are then intelligently converted into *exploratory queries for external scientific literature*. By establishing this seamless link of 'internal reflection $\rightarrow$ active knowledge seeking $\rightarrow$ external knowledge injection', we empower the evolutionary algorithm with an unprecedented capability: when trapped in a local optimum, it can, like a human researcher, consult literature with a specific goal to acquire novel ideas, thus achieving knowledge-driven innovation.

❸ *State-of-the-Art Performance and In-depth Analysis through Rigorous Empirical Validation.* We provide extensive empirical evidence for the superiority of KA-EAD. Our results demonstrate that the framework not only dramatically enhances classical metaheuristics (e.g., achieving a **30.8%** performance improvement for Genetic Algorithm on TSP50) but also significantly optimizes complex, SOTA Neural Combinatorial Optimization (NCO) solvers, for instance, slashing the optimality gap of the LEHD solver on TSP1000 from **27.70%** to a mere **10.71%**.

## 2 RELATED WORK

▶ **Evolutionary Computation for Algorithm Design.** Evolutionary Computation (EC) encompasses a family of population-based metaheuristics inspired by biological evolution, which have long been applied to automated algorithm design and hyper-heuristic discovery Eiben & Smith (2015). Genetic Programming (GP), a prominent EC branch, directly evolves programs or executable structures, making it a natural fit for generating heuristics or even complete algorithms (Koza, 1992; Poli et al., 2008). GP has been utilized to design dispatching rules for scheduling (Koza, 1992), routing heuristics (Mei et al., 2016), and components for metaheuristics. Traditional hyper-heuristics often search a space of predefined heuristic components or select from a fixed set of low-level heuristics

(Burke et al., 2019). While successful in various domains, these approaches can be limited by the human effort required to define the initial set of building blocks or search operators, potentially constraining the novelty of the discovered solutions. *Our work extends this lineage by leveraging LLMs to define and manipulate the individuals (algorithmic heuristics) within an evolutionary framework, thereby vastly expanding the search space beyond manually predefined components.*

▶ **Large Language Models (LLMs) for Algorithm Design.** The advent of LLMs with strong code generation capabilities has opened new avenues for automating aspects of algorithm design (Chen et al., 2021a; Li et al., 2022a; Achiam et al., 2023). LLMs can translate natural language descriptions into code, implement known algorithms from specifications, and even assist in debugging or suggesting algorithmic improvements (Nye et al., 2021; Austin et al., 2021). Recent efforts have begun to explore LLMs not just as code implementers but as active participants in the design process itself. Some works focus on using LLMs for program synthesis under constraints or for few-shot generation of algorithmic solutions (Scholak et al., 2021; Romera-Paredes et al., 2024). More directly related to our work are approaches that integrate LLMs with iterative search or optimization frameworks. FunSearch (Romera-Paredes et al., 2024) demonstrated that an LLM, acting as a sophisticated mutation operator within an evolutionary search, can discover novel and effective functions in mathematical domains by evolving programs. Evolution of Heuristics (EoH) (Liu et al., 2024) and ReEvo Ye et al. (2024) further pushed this boundary by explicitly incorporating LLM-driven 'reflection' or co-evolving natural language 'thoughts' with code to guide the heuristic discovery process. These methods highlight the potential of LLMs to move beyond mere implementation towards creative problem-solving in algorithmic contexts. However, these approaches primarily rely on the LLM's internalized knowledge. *Our work seeks to overcome this limitation by dynamically grounding the LLM's generative and reflective processes in external, domain-specific scientific literature.*

## 3 METHOD

Our Knowledge-Augmented Evolutionary Algorithm Design (KA-EAD) framework introduces a novel paradigm for automated algorithm discovery. Large Language Models (LLMs) act as intelligent generative and mutative operators within an evolutionary computation loop, dynamically guided by insights retrieved from a curated, domain-specific knowledge base. An overview of the KA-EAD framework, illustrating its two main components—***Knowledge Base Construction*** and ***Evolutionary Framework***—is presented in Figure 1.

### 3.1 PROBLEM FORMULATION

We address the challenge of designing high-performance algorithmic components, particularly heuristics for NP-hard optimization problems. Let $\mathcal{P}$ represent a specific computational problem, such as designing the `pick_move` heuristic within an Ant Colony Optimization (ACO) framework for the Traveling Salesman Problem (TSP). The solution space $\mathcal{S}$ comprises all syntactically valid and semantically plausible code implementations for this target component. Our objective is to discover an optimal implementation $s^* \in \mathcal{S}$ that minimizes an objective function $f : \mathcal{S} \to \mathbb{R}$. This function $f(s)$ quantifies the quality of a solution $s$ (represented by its code $c$), typically measured by its performance (e.g., average tour length achieved by the ACO system using heuristic $s$) on a set of benchmark problem instances, evaluated within an execution environment $\mathcal{E}$.

$$s^* = \underset{s \in \mathcal{S}}{\operatorname{argmin}} f(s) \tag{1}$$

The inherent complexity and vastness of $\mathcal{S}$ necessitate intelligent search strategies beyond random exploration or purely human-driven design.

### 3.2 KNOWLEDGE BASE CONSTRUCTION AND AUGMENTATION

A cornerstone of KA-EAD, depicted on the left of Figure 1, is the construction and utilization of a rich, domain-specific knowledge base $\mathcal{K}$. This repository serves as an external, verifiable source of information, empowering the LLM with targeted insights throughout the evolutionary process.

▶ **Literature Corpus Curation and Semantic Filtering ($C_{\text{lit}}$).** We begin by assembling a corpus of relevant scientific literature $C_{\text{lit}} = \{d_1, d_2, \ldots, d_N\}$. An initial set of candidate documents is

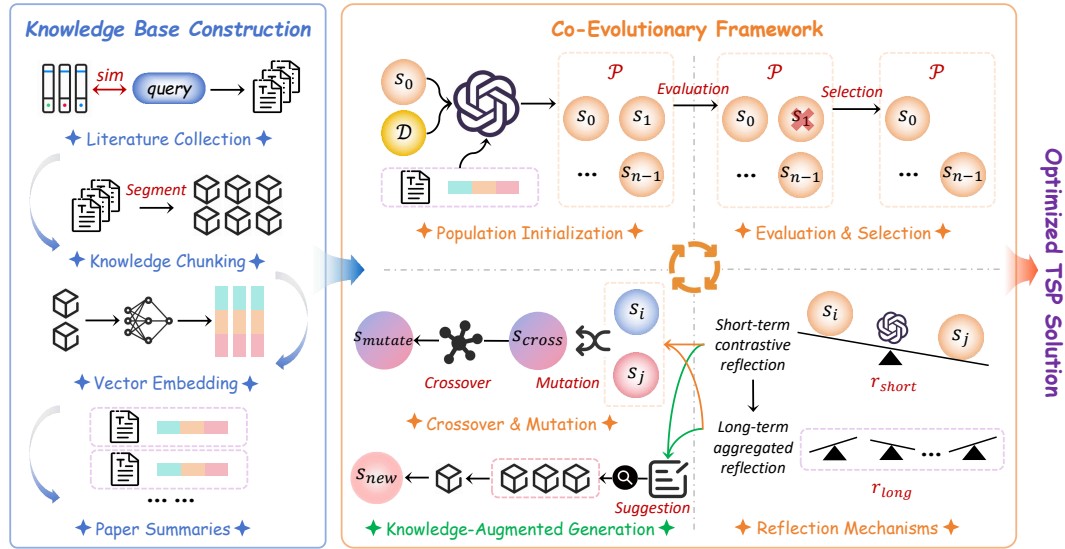

Figure 1: Overview of the Knowledge-Augmented Evolutionary Algorithm Design (KA-EAD) framework. **Left: Knowledge Base Construction** involves collecting and filtering literature, segmenting it into chunks, and embedding these chunks for retrieval. Summaries ($\mathcal{D}$) may also be generated. **Right: Co-Evolutionary Framework** starts with LLM-driven *Population Initialization* for problem $\mathcal{P}$, using summaries $\mathcal{D}$. The loop includes *Evaluation & Selection*, standard *Crossover & Mutation* ($s_{\text{cross}}, s_{\text{mutate}}$), and *Reflection Mechanisms* ($r_{\text{short}}, r_{\text{long}}$) analyzing solutions $s_i, s_j$. Reflections and a focused suggestion ($Q$) guide *Knowledge-Augmented Generation*, where the LLM uses retrieved knowledge to create new solutions $s_{\text{new}}$.

retrieved from academic databases using keywords pertinent to the problem $\mathcal{P}$ (e.g., 'Traveling Salesman Problem', 'Ant Colony Optimization', 'heuristic design'). To refine this set, we employ a sentence embedding model $E_{\text{sent}}$ (e.g., Sentence Transformers' `all-MiniLM-L6-v2`) to generate vector representations for the abstract of each candidate document $d_i$ and for a language description of the problem $\mathcal{P}$. Documents whose abstract embeddings exhibit a cosine similarity score above a predefined threshold $\theta_{\text{corpus}}$ with the problem description embedding are selected for $C_{\text{lit}}$:

$$C_{\text{lit}} = \{d_i \mid \text{sim}(E_{\text{sent}}(\text{abstract}(d_i)), E_{\text{sent}}(\text{desc}(\mathcal{P}))) > \theta_{\text{corpus}}\}. \tag{2}$$

This ensures semantic alignment with the target algorithmic design task.

▶ **Knowledge Chunking, Embedding, and Indexing** ($\mathcal{K}_{\textbf{chunk}}, \mathcal{I}_{\mathcal{K}}$). The full textual content $T_i$ of each document $d_i \in C_{\text{lit}}$ is extracted. To facilitate fine-grained retrieval and manage LLM context windows, each $T_i$ is segmented into smaller, semantically coherent 'knowledge chunks' $\kappa_{i,j}$ using a tokenizer $Tok$ (e.g., from `all-MiniLM-L6-v2`), constraining each chunk's token length to approximately $L_{\text{chunk}}$ tokens (e.g., 512). The union of all such chunks forms the core retrievable knowledge base $\mathcal{K}_{\text{chunk}} = \bigcup_{i,j}\{\kappa_{i,j}\}$. Each knowledge chunk $\kappa \in \mathcal{K}_{\text{chunk}}$ is then transformed into a dense vector representation $v_\kappa \in \mathbb{R}^d$ using an embedding model $E_{\text{chunk}}$ (e.g., `all-MiniLM-L6-v2`): $v_\kappa = E_{\text{chunk}}(\kappa)$. The resulting set of embedding vectors $\{v_\kappa\}$ is organized into a searchable vector index $\mathcal{I}_{\mathcal{K}}$ using FAISS, enabling rapid $k$-nearest neighbor searches. This indexed knowledge forms the operational retrieval system.

### 3.3 Knowledge-Augmented Evolutionary Algorithm

The Knowledge-Augmented Evolutionary Algorithm (KA-EA) framework, detailed in the right panel of Figure 1, iteratively refines a population of candidate algorithmic solutions $Pop_t = \{s_1, s_2, \ldots, s_M\}$ at evolutionary iteration $t$, where $M$ is the population size. Each solution $s_k$ is primarily defined by its source code $c_k$.

▶ **Initial Population Generation** ($Pop_0$). The evolutionary search commences with the generation of an initial population $Pop_0$ by a generator LLM, $LLM_{\text{gen}}$ (e.g., gpt-3.5-turbo). To provide ini-

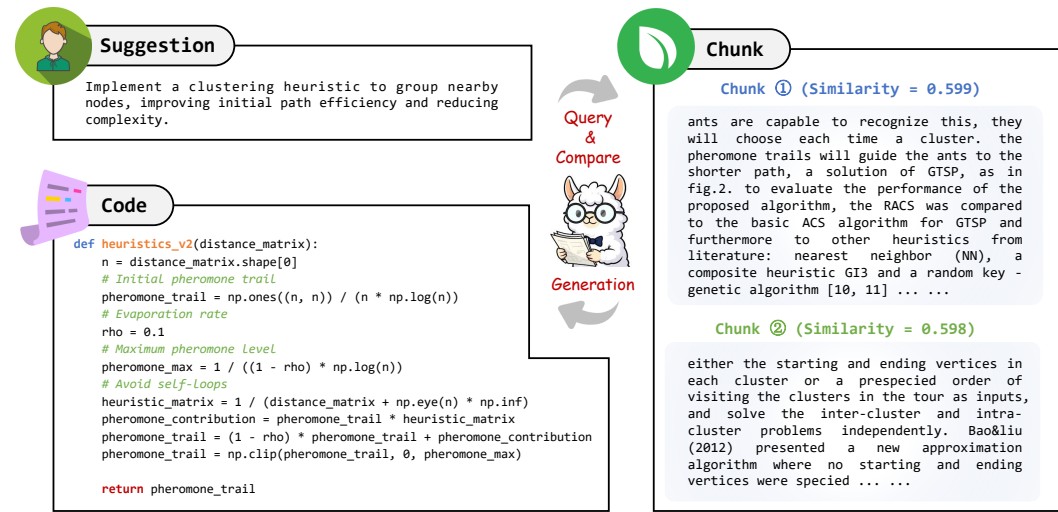

Figure 2: Detailed overview of the Retrieval-Augmented Generation (RAG) operator. The process begins with the large language model generating a random suggestion, followed by querying the knowledge base to retrieve the two most similar content chunks. These chunks serve as additional context, enhancing the model's ability to produce optimized algorithms.

tial grounding, a comprehensive summary of domain-specific knowledge, $Summ(\mathcal{K}_{\text{chunk}})$, derived from $\mathcal{K}_{\text{chunk}}$ (and potentially illustrated as $\mathcal{D}$ in Figure 1), is incorporated into the initial prompt. Let $\pi_{\text{seed}}$ be a seed prompt outlining problem $\mathcal{P}$ and the desired solution structure (e.g., Python function signature), and $\rho_{LT,0}$ be the initial (typically empty) long-term reflection string. The full initial prompt is $\text{prompt}_{\text{init}} = \text{SystemPrompt}_{\text{gen}} \oplus \text{UserPrompt}(\pi_{\text{seed}}, \rho_{LT,0}, Summ(\mathcal{K}_{\text{chunk}}))$, where $\oplus$ denotes concatenation. $LLM_{\text{gen}}$ then samples $M$ distinct code snippets $\{c_1, \ldots, c_M\}$ based on $\text{prompt}_{\text{init}}$, often with a slightly elevated temperature $\tau_{\text{init}}$ to encourage diversity: $c_k \sim LLM_{\text{gen}}(\text{prompt}_{\text{init}}, \text{temperature} = \tau_{\text{init}})$. Each $c_k$ is instantiated as $s_k \in Pop_0$, and its fitness $f(s_k)$ is determined by execution in environment $\mathcal{E}$.

▶ **Evolutionary Operators and Loop.** For each subsequent iteration $t > 0$, KA-EA employs a sequence of core operators as shown in Figure 1. Firstly, ***Selection*** identifies parent solutions $Pop'_{t-1} \subseteq Pop_{t-1}$ based on fitness $f(s)$, typically via tournament selection. Secondly, the ***Reflection*** mechanism, utilizing an LLM denoted $LLM_{\text{reflect}}$, analyzes pairs of high-performing ($s_{\text{better}}$) and low-performing ($s_{\text{worse}}$) solutions from $Pop'_{t-1}$. This process yields natural language short-term reflections $\rho_{ST,t}$ concerning effective versus ineffective algorithmic constructs: $\rho_{ST,t} = LLM_{\text{reflect}}(\text{prompt}_{\text{reflect}}(c_{\text{better}}, c_{\text{worse}}, f(s_{\text{better}}), f(s_{\text{worse}})))$. The long-term reflection $\rho_{LT,t}$ is then updated by accumulating new insights from $\rho_{ST,t}$ with the previous $\rho_{LT,t-1}$. Thirdly, ***Standard Variation Operators***, such as crossover and mutation, are applied. These conventional EC operators, which might involve LLM-prompted code combination or modification, act on solutions in $Pop'_{t-1}$ and can be guided by $\rho_{ST,t}$ to produce a set of offspring $Pop_{\text{var},t}$.

The fourth and pivotal operator is the ***Dynamic Knowledge-Augmented Generation Operator (RAG-Operator)***, which dynamically infuses external knowledge through a three-stage process. A detailed schematic of this RAG-Operator, illustrating the flow from an initial suggestion to knowledge retrieval and subsequent augmented code generation, is provided in Figure 2. The stages are:

❶ ***Query Formulation ($q_{RAG}$):*** An LLM, $LLM_{\text{query}}$, synthesizes a focused query $q_{\text{RAG}}$ for the knowledge base. This query is conditioned on the current reflections $\rho_{ST,t}$, $\rho_{LT,t}$, and a specific problem_focus (e.g., 'how to improve exploration in ACO for TSP'), as defined by $q_{\text{RAG}} = LLM_{\text{query}}(\text{prompt}_{\text{query\_form}}(\rho_{ST,t}, \rho_{LT,t}, \text{problem\_focus}))$. This query or suggestion is analogous to the 'Suggestion' input shown in Figure 2.

❷ ***Knowledge Retrieval ($\mathcal{K}_{retrieved}$):*** The formulated query $q_{\text{RAG}}$ is embedded into a vector $v_q = E_{\text{chunk}}(q_{\text{RAG}})$. This vector is then used to perform a $k_{\text{ret}}$-nearest neighbor search within the

indexed knowledge base $\mathcal{I}_\mathcal{K}$, yielding a set of relevant knowledge chunks $\mathcal{K}_{\text{retrieved}} = \{\kappa_j \mid \kappa_j = \text{NN}_j(\mathcal{I}_\mathcal{K}, v_q), j = 1, \ldots, k_{\text{ret}}\}$. These are the 'content chunks' depicted in Figure 2.

❸ *Augmented Generation:* A highly contextualized prompt, $\text{prompt}_{\text{RAG}} = \text{SystemPrompt}_{\text{gen}} \oplus \text{UserPrompt}(\text{task\_spec}, \mathcal{K}_{\text{retrieved}}, \text{improvement\_suggestion})$, is constructed. This prompt integrates system instructions, the specific task description (e.g., Python function signature for `pick_move` including type and tensor constraints), the retrieved knowledge $\mathcal{K}_{\text{retrieved}}$, and an improvement suggestion (which could be $q_{\text{RAG}}$ itself). $LLM_{\text{gen}}$ then generates new candidate solutions $Pop_{\text{RAG},t}$, similar to the 'Code' output in Figure 2. An iterative refinement loop is employed if the initial code fails syntactic or semantic checks, by re-prompting the LLM with error feedback to ensure code correctness.

▶ **Evaluation, Population Update, and Termination.** All newly generated solutions ($s \in Pop_{\text{var},t} \cup Pop_{\text{RAG},t}$) are compiled and evaluated in $\mathcal{E}$ to obtain fitness $f(s)$; the function evaluation counter $FE$ is incremented. The next population $Pop_t$ is formed by selecting the $M$ best individuals from $Pop_{t-1} \cup Pop_{\text{var},t} \cup Pop_{\text{RAG},t}$, often using elitism. The process continues until a termination criterion ($FE \geq FE_{\max}$, fitness convergence, or time limit) is met. The best solution $s^*$ found is returned.

This comprehensive framework allows KA-EAD to leverage the broad generative capabilities of LLMs while grounding their outputs in specific, relevant, and verifiable knowledge from scientific literature, thereby fostering the discovery of novel and effective algorithmic solutions.

## 4 EXPERIMENTS

▶ **Datasets.** For the more complex NCO solvers, we used larger-scale benchmarks. Our evaluation spans both classical and modern optimization paradigms. For classical algorithms—Genetic Algorithm (GA), Ant Colony Optimization (ACO), and Kernighan-Lin Heuristic with Local Search (KGLS)—we generated TSP instances with node coordinates uniformly distributed in $[0, 1)^2$. The training set comprised 5 instances of size 50, providing a focused environment for heuristic evolution. The test set included 64 instances for each of three sizes (20, 50, and 100) to assess generalization. For the more complex NCO solvers, we used larger-scale benchmarks. For Path-Optimization with Multi-Objective (POMO), we used instances of sizes 200, 500, and 1000. For the Learning-based Euclidean Heuristic Decomposition (LEHD) algorithm, we utilized the publicly available TSP benchmark dataset from the CIAM-Group on instances of the same large sizes. This dual-dataset strategy ensures our evaluation is robust across different problem scales and types.

▶ **Implementation Details.** For all LLM-driven operations within KA-EAD and the baselines, we consistently used the `gpt-4o-mini` model to ensure a fair comparison of the frameworks' methodologies. The evolutionary process, unless otherwise noted, used an initial population of 30, an evolving population of 10, and was run for a maximum of 100 function evaluations (`max_fe`). Specific key configurations include: For **ACO**, the population size was set to 30 for both initialization and evolution. For **NCO solvers**, the heuristic evolution was performed on size 500 instances, with the neural models trained for 50 epochs. For complete reproducibility, a comprehensive description of all hyperparameters, specific algorithm configurations, and the LLM prompt structures (see exemplars in **Appendix 4**) are provided in **Appendix 1**.

▶ **Baselines.** Our evaluation framework is built upon a comprehensive set of baselines to rigorously assess the contributions of KA-EAD.

❶ *Classical Solvers:* Standard implementations of GA, ACO, and KGLS serve as our foundational baselines, representing widely-accepted, human-designed algorithmic structures.

❷ *NCO Solvers:* We include POMO and LEHD as representatives of modern, learning-based approaches to combinatorial optimization, providing a testbed for KA-EAD's ability to enhance complex neural components.

❸ *LLM-based Heuristic Generation:* To isolate the benefit of external knowledge retrieval, we compare directly against EOH (Liu et al., 2024) and ReEvo (Ye et al., 2024). These methods represent the SOTA in using LLMs for heuristic discovery, relying solely on the model's internalized knowledge and reflection mechanisms.

Table 1: Performance of different heuristic methods on various algorithms for optimizing the solution to TSP. All reported objective values and time ratios are the average of three independent runs.

| Type | TSP20 | | | TSP50 | | | TSP100 | | |
|---|---|---|---|---|---|---|---|---|---|
| | Obj ↓ | Gap (%) ↑ | Time ratio ↓ | Obj ↓ | Gap (%) ↑ | Time ratio ↓ | Obj ↓ | Gap (%) ↑ | Time ratio ↓ |
| GA | 6.1 | 0.0 | 1.3 | 18.2 | 0.0 | 1.625 | 40.8 | 0.0 | 1.1 |
| GA + EOH | 6.0 | 1.6 | 1.0 | 17.8 | 2.2 | 1.0 | 40.5 | 0.7 | 1.0 |
| GA + ReEvo | 6.0 | 1.6 | 1.0 | 17.9 | 1.6 | 1.0 | 40.6 | 0.5 | 1.0 |
| GA + KA-EAD | 4.8 | 21.3 | 1.9 | 12.6 | 30.8 | 1.5 | 30.0 | 26.5 | 1.4 |
| ACO | 3.9 | 0.0 | 0.7 | 5.9 | 0.0 | 0.8 | 8.7 | 0.0 | 0.7 |
| ACO + EOH | 3.9 | 0.0 | 1.4 | 5.9 | 0.0 | 1.2 | 8.5 | 2.3 | 1.4 |
| ACO + ReEvo | 3.9 | 0.0 | 1.0 | 5.9 | 0.0 | 1.0 | 8.5 | 2.3 | 1.0 |
| ACO + KA-EAD | 3.4 | 12.8 | 1.1 | 5.4 | 8.5 | 1.1 | 7.9 | 9.2 | 1.3 |
| KGLS | 4.4 | 0.0 | 0.7 | 6.7 | 0.0 | 0.7 | 9.3 | 0.0 | 1.3 |
| KGLS + EOH | 4.4 | 0.0 | 0.9 | 6.8 | -1.5 | 0.9 | 9.2 | 0.1 | 1.4 |
| KGLS + ReEvo | 4.4 | 0.0 | 1.0 | 6.8 | -1.5 | 1.0 | 9.3 | 0.0 | 1.0 |
| KGLS + KA-EAD | 3.9 | 11.4 | 2.5 | 5.7 | 14.9 | 1.5 | 7.8 | 16.1 | 0.6 |

Table 2: Evaluation results for NCO solvers with and without different attention-reshaping heuristics. All reported objective values and optimality gaps are the average of three independent runs.

| Method | n = 200 | | n = 500 | | n = 1000 | |
|---|---|---|---|---|---|---|
| | Obj ↓ | Opt. gap (%) ↓ | Obj ↓ | Opt. gap (%) ↓ | Obj ↓ | Opt. gap (%) ↓ |
| POMO | 15.35 | 43.59 | 25.58 | 54.84 | 38.79 | 67.63 |
| POMO + EOH | 15.68 | 46.67 | 25.36 | 53.51 | 38.79 | 67.63 |
| POMO + ReEvo | 15.39 | 43.96 | 25.43 | 53.93 | 38.79 | 67.63 |
| POMO + KA-EAD | 14.68 | 37.32 | 23.36 | 41.40 | 34.86 | 50.64 |
| LEHD | 14.79 | 38.35 | 20.78 | 25.78 | 29.55 | 27.70 |
| LEHD + DAR | 14.88 | 39.19 | 19.79 | 19.79 | 27.85 | 20.35 |
| LEHD + ReEvo | 14.77 | 38.16 | 19.78 | 19.73 | 27.62 | 19.36 |
| LEHD + KA-EAD | 13.33 | 24.69 | 18.52 | 12.10 | 25.62 | 10.71 |

## 4.1 RESULTS AND ANALYSIS

We present the performance results of KA-EAD on classical and NCO solvers for the Traveling Salesman Problem (TSP), followed by ablation studies and qualitative analysis to dissect the framework's components and behavior. To demonstrate the general applicability, we also report results on other NP-hard problems.

▶ **Performance on Classical TSP Solvers.** Table 1 summarizes the performance of GA, ACO, and KGLS when augmented with different heuristic generation methods. The results unequivocally demonstrate the superiority of our knowledge-augmented approach. Across all three classical algorithms and all problem sizes, ***KA-EAD consistently achieves the lowest objective values (Obj ↓) and, consequently, the largest improvement gaps (Gap (%) ↑) over the vanilla baselines.*** For instance, when applied to GA on TSP50, KA-EAD yields an objective of 12.6 and an improvement gap of 30.8%, substantially outperforming EOH (17.8 Obj, 2.2% Gap) and ReEvo (17.9 Obj, 1.6% Gap). Similar significant gains are observed for ACO and KGLS. This highlights that grounding the LLM's search in scientific literature unlocks performance levels unattainable by methods relying only on internal knowledge.

▶ **Enhancement of Neural Combinatorial Optimization Solvers.** Table 2 details the results of applying KA-EAD to improve components within NCO solvers on large-scale TSP instances. KA-EAD again demonstrates remarkable efficacy. For both POMO and LEHD, ***integrating KA-EAD leads to the lowest objective values and the smallest optimality gaps (Opt. gap (%) ↓) across all tested problem sizes.*** A standout result is with LEHD on TSP1000 instances, where KA-EAD reduces the optimality gap from 27.70% (vanilla LEHD) to a mere 10.71%. This improvement significantly surpasses other methods, including the specialized DAR baseline. These findings underscore KA-EAD's capability to effectively guide the design of sophisticated components within complex learning-based systems.

▶ **Ablation and Component Analysis.** To understand the contributions of KA-EAD's core components, we conducted extensive ablation studies. As shown in the left panels of Figure 3, the

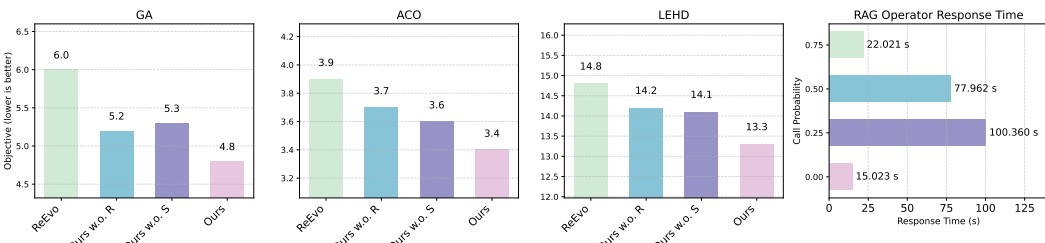

Figure 3: Ablation and Efficiency Analysis of KA-EAD. **Left**: Ablation study on GA, ACO, and LEHD. The full method ('Ours') outperforms variants without Reflection ('w.o. R') or Scientific Knowledge ('w.o. S'), demonstrating the synergy of its components. **Right**: Average response time of the RAG operator. Higher call probabilities lead to faster generation, suggesting that knowledge infusion improves search efficiency.

full KA-EAD framework consistently outperforms ablated versions that lack either the reflection mechanism ('Ours w.o. R') or the scientific knowledge retrieval ('Ours w.o. S'). This performance drop underscores the critical and synergistic roles of both internal reflection and external knowledge infusion. The right panel of Figure 3 also reveals an interesting efficiency aspect: more frequent knowledge retrieval can surprisingly reduce the average code generation time. A more detailed breakdown of these ablation results can be found in **Appendix 2**.

▶ **Generalization to Other NP-Hard Problems.** To demonstrate the broader applicability and robustness of KA-EAD, we evaluated its performance on a diverse suite of 15 well-known NP-hard problems, comparing it against LLM-based baselines, EOH and ReEvo. The detailed results are presented in Table 3.

The data unequivocally shows the superiority of our knowledge-augmented approach. KA-EAD ("Ours") consistently and substantially outperforms both baselines across nearly all tasks. The performance gap is particularly pronounced in complex domains requiring specialized heuristics. For instance, in *Flow Shop Scheduling (FSS)*, our method achieves an 84.83% performance ratio on the test set, whereas ReEvo and EOH only reach 53.92% and 30.65%, respectively. Similarly, for the *Generalized Assignment Problem (GAP)*, KA-EAD surpasses the next-best baseline by nearly 15 percentage points. This consistent outperformance across a wide variety of problem structures underscores the critical advantage of dynamically grounding the heuristic search in external scientific literature, rather than relying solely on a model's static, internalized knowledge.

Table 3: Detailed performance comparison on a broad set of 15 NP-hard problems. Values represent the performance ratio to known optimal solutions. 'Comp.' denotes composite performance across all datasets, and 'Test' refers to performance on the test set. Full problem names and citations are provided below.

| Task | EOH (%) | | ReEvo (%) | | Ours (%) | |
|---|---|---|---|---|---|---|
| | Comp. | Test | Comp. | Test | Comp. | Test |
| BP1D[a] | 88.37 | 89.21 | 89.40 | 89.04 | **96.03** | **96.36** |
| CStr[b] | 93.16 | 94.50 | 93.16 | 94.50 | **99.87** | **99.75** |
| HRSS[c] | 87.81 | 88.41 | 91.26 | 91.97 | **94.99** | **95.26** |
| OpSS[d] | 75.40 | 72.48 | 78.27 | 78.48 | **90.77** | **90.17** |
| UWLoc[e] | 98.21 | 97.72 | 98.10 | 97.69 | **99.84** | **98.82** |
| UGCut[f] | 80.77 | 82.50 | **93.02** | 93.88 | 91.39 | **94.19** |
| AL[g] | 80.76 | 80.57 | 81.38 | 80.23 | **82.89** | **82.22** |
| CWLoc[h] | 73.52 | 73.59 | 66.97 | 67.09 | **74.07** | **74.17** |
| CDDS[i] | 77.99 | 77.69 | 81.39 | 81.01 | **82.35** | **83.43** |
| CL[j] | 71.93 | 71.54 | 72.88 | 72.48 | **73.32** | **73.94** |
| CSch[k] | 15.12 | 11.17 | 40.60 | 44.57 | **52.07** | **57.87** |
| FSS[l] | 32.83 | 30.65 | 56.79 | 53.92 | **84.92** | **84.83** |
| GAP[m] | 67.09 | 65.75 | 74.46 | 72.83 | **86.49** | **87.38** |
| JSS[n] | 66.14 | 64.89 | 73.14 | 71.86 | **74.39** | **73.78** |
| SP[o] | 30.50 | 23.33 | 47.45 | 40.00 | **76.27** | **73.33** |

**Task Definitions**: **a**: BP1D (Falkenauer, 1996); **b**: CStr (Anken & Beasley, 2012); **c**: HRSS (Chakhlevitch & Glass, 2009); **d**: OpSS (Taillard, 1993); **e**: UWLoc (Beasley, 1988); **f**: UGCut (Beasley, 1985); **g**: AL (Beasley et al., 2000); **h**: CWLoc (Beasley, 1988); **i**: CDDS (Biskup & Feldmann, 2001); **j**: CL (Bischoff & Ratcliff, 1995); **k**: CSch (Beasley & Cao, 1996); **l**: FSS (Taillard, 1993); **m**: GAP (Osman, 1995); **n**: JSS (Taillard, 1993); **o**: SP (Chu & Beasley, 1998).

▶ **Qualitative Analysis and Evolutionary Trajectory.** The quantitative superiority of KA-EAD is rooted in its ability to generate higher-quality heuristics and sustain a more effective search. Qualitatively, this is evident in the final generated code; for instance, the final `pick_move` function generated by KA-EAD for ACO (presented in Figure 8 in the **Appendix**) incorporates more nuanced logic compared to baselines, likely reflecting a synthesis of insights from the retrieved literature.

This is further substantiated by the evolutionary trajectory analysis in Figure 4b. The plot reveals that ***KA-EAD exhibits a consistent and sustained improvement trajectory, achieving better objective values over generations.*** In stark contrast, ***the ReEvo baseline stagnates quickly after the initial iterations, its performance plateauing at a suboptimal level.*** This divergence is a typical manifestation of premature convergence, which for ReEvo results directly from its reliance on static, internalized knowledge. Conversely, the dynamic infusion of external knowledge via our RAG-Operator (detailed in Figure 2 and **Appendix 3**) empowers KA-EAD to navigate the search space more effectively, successfully avoiding this pitfall to discover superior solutions.

▶ **Analysis of the RAG Operator's Behavior.** We conducted an in-depth analysis of the RAG operator's behavior to quantify its contribution to the evolutionary process. Figure 4a visually demonstrates the direct impact of the RAG operator's activation probability on performance. A clear trend is observable: as the probability of invoking the RAG operator (p-value) increases, the algorithm not only achieves lower objective values more rapidly, but the distribution of solution quality also becomes more concentrated. For instance, with p=0.75, the algorithm not only reaches the lowest median objective value but also exhibits significantly less variance in later iterations compared to scenarios with lower probabilities or with p=0 (no knowledge infusion). This provides strong evidence that frequently grounding the search process in external scientific knowledge is a key mechanism for enhancing both solution quality and search efficiency.

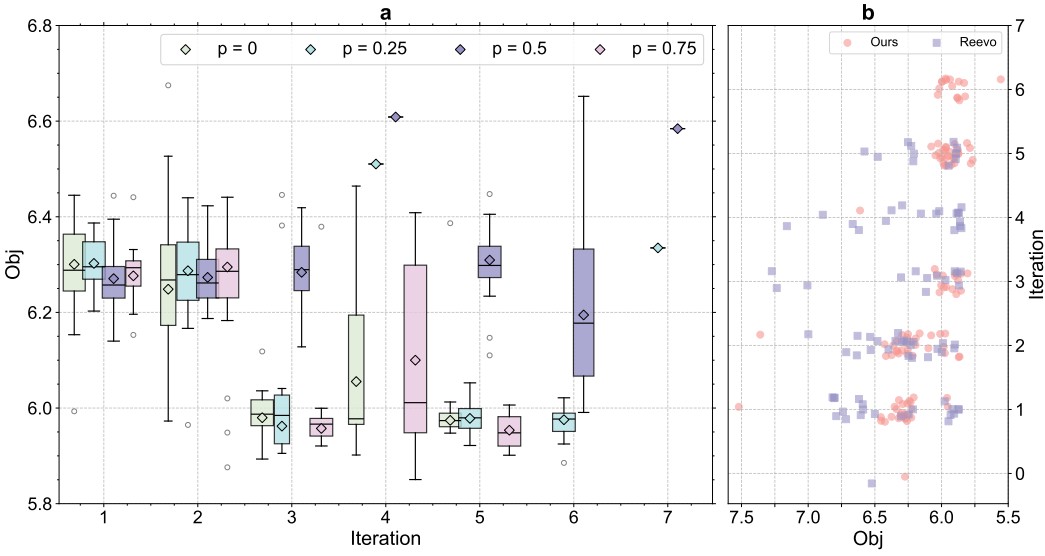

Figure 4: Analysis of the RAG operator's impact and evolutionary trajectory. This figure illustrates the critical role of the Knowledge-Augmented Generation (RAG) operator in the optimization process. **Left (a)** shows the distribution of objective values (Obj) across evolutionary iterations under different activation probabilities (p) for the RAG operator. Higher activation probabilities (e.g., p=0.75) consistently lead to lower (better) and more stable objective values. This indicates that frequently grounding the search in external scientific knowledge via the RAG operator significantly helps the evolutionary process escape local optima and discover higher-quality solutions. **Right (b)** compares the evolutionary trajectories of our proposed KA-EAD method ('Ours', red circles ) and the baseline ReEvo ('Reevo', blue squares ) on the ACO algorithm.

## 5 CONCLUSION

This paper introduced KA-EAD, a framework that enables LLMs to transcend their static knowledge boundaries by actively integrating insights from scientific literature. It transforms an LLM's internal reflections into targeted queries for external knowledge, creating a powerful search dynamic that emulates the cycle of scientific inquiry. By demonstrating that this synergy consistently discovers superior algorithmic solutions, KA-EAD establishes a new paradigm for building AI systems that can learn from, and build upon, the cumulative progress of human scientific knowledge.

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

# Appendix
# Fusing LLMs with Scientific Literature for Heuristic Discovery

This appendix provides supplementary material to accompany the main paper. It is organized as follows:

- **Section A:** A comprehensive description of the experimental setup, including datasets, algorithm configurations, and evaluation metrics.
- **Section B:** Additional experimental results, including detailed ablation studies.
- **Section C:** Pseudocode detailing the inner workings of the RAG operator.
- **Section D:** Exemplar prompt structures used to guide the Large Language Models.
- **Section E:** Use of LLMs.

## A    EXPERIMENTAL DETAILS

This section provides a comprehensive overview of the experimental setup, detailing the datasets, baseline algorithms, evaluation metrics, and specific configurations employed in our study, corresponding to the main paper's experiments.

### A.1    DATASETS AND PROBLEM SPECIFICATION

For the evaluation of classical algorithms, namely Genetic Algorithm (GA), Ant Colony Optimization (ACO), and Kernighan-Lin Heuristic with Local Search (KGLS), we generated TSP instances where node coordinates were uniformly distributed in the interval $[0, 1)^2$ using the NumPy library. The training set for these algorithms consisted of 5 instances, each with 50 nodes. For testing, we utilized 64 instances for each of the problem sizes: 20, 50, and 100 nodes.

For Neural Combinatorial Optimization (NCO) solvers, specific datasets were used. Path-Optimization with Multi-Objective (POMO) was evaluated on problem sizes of 200, 500, and 1000 nodes. Its training dataset contained 10 instances per problem size, while both validation and test sets comprised 64 instances per size. For the Learning-based Euclidean Heuristic Decomposition (LEHD) algorithm, we employed the publicly available TSP benchmark dataset from the CIAM-Group code library, focusing on test instances of 200, 500, and 1000 nodes, where node coordinates are also uniformly distributed in $[0, 1)^2$.

### A.2    ALGORITHM CONFIGURATIONS AND SETTINGS

A consistent Large Language Model (LLM), `gpt-4o-mini`, was utilized for all LLM operations within our KA-EAD framework—encompassing initial population generation, reflection, query formulation, and code generation—as well as for implementing the LLM-based baselines EOH (Liu et al., 2024) and ReEvo (Ye et al., 2024). Unless otherwise specified, evolutionary experiments shared general parameters: an initial population of 30 individuals, an evolving population of 10 individuals, and a maximum of 100 function evaluations (`max_fe`).

#### A.2.1    CLASSICAL SOLVERS AND LLM-BASED AUGMENTATION

We used classic solver baselines - GA, ACO, and KGLS - as the base comparison, and enhanced them by integrating heuristics designed or improved by KA-EAD, EOH, and ReEvo.

- For **GA**, the LLM-generated components are crossover heuristics. If not generated by LLM, basic parameters such as 'elite_rate=0.2', 'n_iter=80', and 'n_pop=80' are applied.

- For **ACO**, the main target of LLM-based enhancement is the `pick_move` heuristic. The initial and evolution population sizes are both set to 30. Other standard ACO parameters include 'n_ants=30', 'evaporation_rate=0.1', and alpha/beta factors both set to 1, used when not implicitly handled by the LLM-generated heuristic.

- For **KGLS**, the LLM targets the core heuristic algorithm. Basic KGLS parameters include 'n_starts=10', 'n_perturbations=30', and an iteration limit of 1000.

### A.2.2 NCO Solvers and LLM-Based Component Enhancement

For modern learning-based NCO solvers (POMO and LEHD), the LLM-based approach aims to incorporate the training process into the evaluation operator of the outer evolutionary algorithm.

- For **POMO**, the evolutionary process for component enhancement uses 8 evolution populations, and its neural model is trained for 50 epochs on a problem of size 500. Neural components were initialized with pre-trained weights from the official codebase.

- For **LEHD**, the neural network is similarly trained for 50 epochs, followed by an evolutionary search with heuristic improvements on a TSP instance with 500 nodes. As mentioned in the main paper (Table 2), we also include the Differentiable Attention Reorganization (DAR) method as a specialized baseline.

### A.3 Targeted Algorithmic Components

The specific algorithmic components targeted by KA-EAD and other LLM-based methods for design or enhancement varied by algorithm:

- **For ACO:** The `pick_move` heuristic function.
- **For GA:** The `crossover` function.
- **For KGLS:** The core heuristic function.
- **For POMO and LEHD:** The training process is integrated into the evaluation operator of the outer algorithm.

### A.4 Evaluation Metrics

Algorithm performance was assessed using several metrics, as defined in the main paper. The primary metric was the **Objective value** ($\downarrow$). We also computed the **Improvement Gap (%)** $\uparrow$ against vanilla baselines and the **Optimality Gap (%)** $\downarrow$ for NCO solvers relative to known optimal or best-known solutions. The **Time ratio** $\downarrow$ reported in Table 1 indicates the computation time relative to the ReEvo baseline.

### A.5 Execution Environment

All experiments were conducted on a machine with an AMD Ryzen 9 7940HX CPU, 32GB RAM, and an NVIDIA RTX 4060 Laptop GPU (8GB VRAM). The software stack included Windows 11, Python 3.12, and PyTorch 2.7.0. LLM access was managed via the agicto platform.

## B Additional Results and Ablation Studies

### B.1 Ablation Analysis of KA-EAD Components

To rigorously evaluate the contribution of each key component in our framework, we conducted comprehensive ablation studies. Figure 5 shows a detailed ablation analysis on NCO solvers, corresponding to the summarized discussion in the main paper. The results consistently demonstrate that the full KA-EAD framework ('Ours') achieves the lowest objective values. The performance degradation in the ablated versions—without the Reflection mechanism ('Ours w.o. R') and without the scientific knowledge retrieval ('Ours w.o. S')—underscores the critical and synergistic contributions of these components.

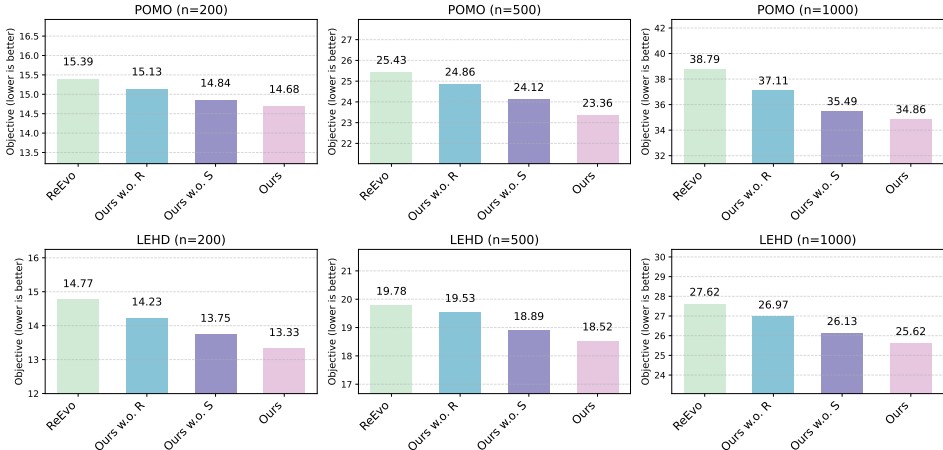

Figure 5: **Ablation Analysis of KA-EAD on NCO Solvers (Detailed View).** This figure presents a comprehensive ablation study comparing the performance of ReEvo (baseline) with three variants of our KA-EAD framework on the POMO and LEHD NCO solvers across TSP instances of size 200, 500, and 1000. The variants include our full method ('Ours'), our method without the Reflection mechanism ('Ours w.o. R'), and our method without the scientific knowledge retrieval operator ('Ours w.o. S'). The results consistently demonstrate that the full KA-EAD framework achieves the lowest objective values (lower is better) in all settings. The performance degradation observed in the ablated versions underscores the critical contributions of both reflection and dynamic knowledge retrieval.

## C    ALGORITHMIC DETAIL OF THE RAG OPERATOR

This section provides a focused conceptual description of the Knowledge-Augmented Generation (RAG) Operator, which is central to the KA-EAD framework. A detailed visual representation is provided in the main paper in Figure 2. The operator's procedure is as follows:

1. **Input**: The operator takes as input the current short-term ($\rho_{ST}$) and long-term ($\rho_{LT}$) reflections from the evolutionary process.
2. **Query Formulation**: An LLM is prompted to synthesize these reflections into a concise, forward-looking 'suggestion' or query. This query is designed to seek novel information that addresses current performance bottlenecks or explores new strategic directions.
3. **Knowledge Retrieval**: The generated query is embedded into a vector representation. This vector is used to perform a semantic search (e.g., k-nearest neighbors) against a pre-indexed knowledge base of scientific literature chunks. The top-k most relevant chunks are retrieved.
4. **Augmented Generation**: A new prompt is constructed for the generative LLM. This prompt includes the original task, the retrieved knowledge chunks as context, and the focused suggestion from step 2.
5. **Output**: The LLM generates a new heuristic solution (code), which is now informed by both the internal evolutionary history (reflections) and external scientific knowledge. This knowledge-infused generation aims to produce more sophisticated and effective solutions.

## D    PROMPT STRUCTURES FOR LLM OPERATORS

This section illustrates the prompt structures employed to guide LLMs within the KA-EAD framework. Effective prompting is key to leveraging the LLM's capabilities for structured code generation.

Figure 6: Exemplar prompt structures for the LLM-based Crossover and Mutation operators. The prompts typically include placeholders for user generator instructions, existing code (e.g., worse or better code for crossover, elitist code for mutation), and reflections, followed by a directive to generate improved or mutated code.

### D.1 PROMPTS FOR CROSSOVER AND MUTATION

For standard evolutionary operators like crossover and mutation, we use specific prompt templates that provide context to the LLM. Figure 6 depicts the general structure of these prompts. They typically include placeholders for parent solutions, reflective insights, and a clear directive to generate a new, improved code segment. For these common operators, we adopted prompt strategies similar to those established in prior work, such as ReEvo.

### D.2 PROMPT FOR ALGORITHM-SPECIFIC COMPONENT: ACO `PICK_MOVE` FUNCTION

When targeting a specific, complex algorithmic component like the `pick_move` function in ACO, a highly detailed and domain-specific prompt is crucial. Figure 7 showcases the comprehensive function description provided to the LLM. This prompt specifies the function's purpose, optimization goals, precise definitions of input parameters (including type and tensor shapes), and the expected output format. This level of detail enables the LLM to generate code that is not only syntactically correct but also contextually relevant and functionally effective. The placeholders within these prompts are dynamically populated based on the current state of the evolutionary process.

## E USE OF LLMS.

LLM is the key part of the method in this paper. In addition, we used LLM for writing polish to improve readability.

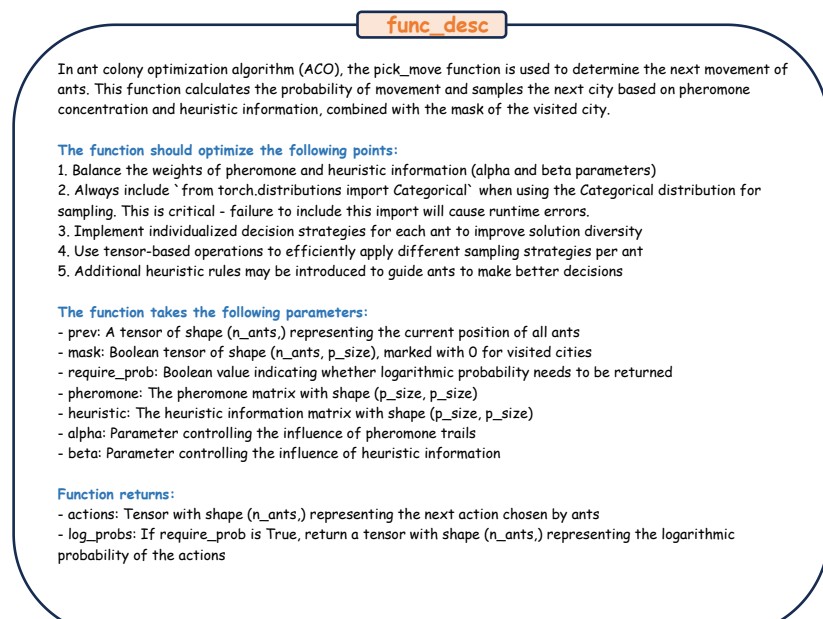

Figure 7: Detailed function description prompt provided to the LLM for the ACO `pick_move` function. This prompt specifies the function's role, optimization goals, precise parameter definitions (name, type, shape, description), and the structure of its return values, enabling the LLM to generate code that is contextually relevant and functionally correct.

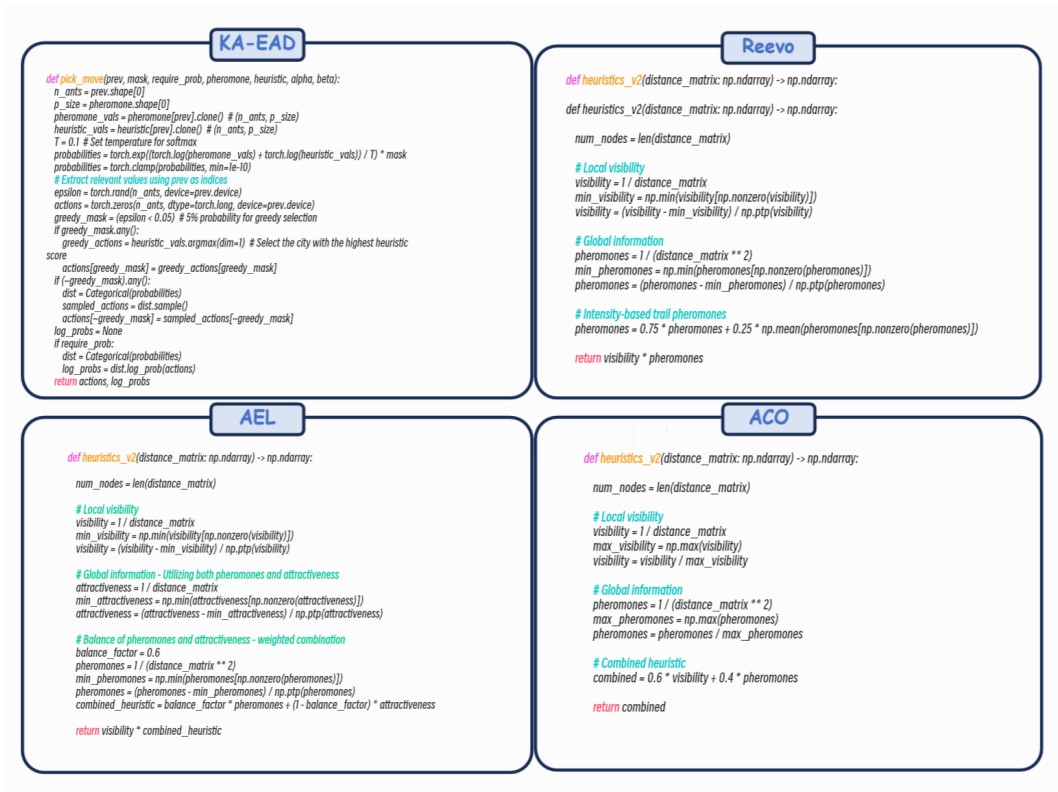

Figure 8: The final code of the aco algorithm is shown. We compared the algorithms generated by KA-EAD, Reevo, and AEL on the ACO algorithm and found that KA-EAD can generate better functions.

