# OpenReview forum: "Fusing LLMs with Scientific Literature for Heuristic Discovery"
_ICLR.cc/2026/Conference — Submitted to ICLR 2026_

### Official Review · Reviewer_YSCm · 2025-10-24

**Soundness:** 2
**Presentation:** 3
**Contribution:** 2
**Rating:** 4
**Confidence:** 3

**Summary:**

The paper proposes a framework where a large language model designs and refines algorithms while dynamically consulting scientific literature for guidance. By embedding retrieval-augmented generation into an evolutionary optimization loop, the model can “learn” from external research and iteratively improve its code.

Honestly, I am not very familiar with NP-hard optimization problems, so my comments mainly focus on the LLM and retrieval aspects of the work. From that perspective, the paper is best viewed as a systems-level integration that embeds a RAG pipeline into an LLM-driven evolutionary search loop, with reflection-driven query formulation. I find this an interesting orchestration, though not a fundamentally new retrieval or learning mechanism. If I have misunderstood any aspects of the work, I welcome the authors to clarify them during the rebuttal phase, and I will adjust my evaluation accordingly.

**Strengths:**

• The authors attempt to tackle a challenging and meaningful problem by leveraging LLMs for NP-hard optimization. As noted in the paper, these problems are pervasive in the real world and often require deep domain expertise and heuristic exploration, which have traditionally been difficult to automate effectively.

• The idea of injecting external, verifiable, and dynamically retrieved scientific knowledge into the evolutionary algorithm design process is conceptually sound and intuitively appealing.

• As reported by the authors, their proposed approach achieves state-of-the-art performance across multiple benchmarks, demonstrating the potential effectiveness of the framework.

**Weaknesses:**

• The paper is heavily engineering-oriented and mostly builds on stacking existing techniques without presenting any notably novel insight. The first claimed contribution essentially overlaps with what RAG is originally designed to address, and the second one feels like a combination of consistency checking and chain-of-thought reflection for hallucination control. Both seem rather trivial for an ICLR submission, which is the main reason I rated it a 4.

• The authors mention that after each generation step, the LLM performs self-reflection and then compresses these reflections into forward-looking suggestions or queries. However, this raises an important issue of balancing additional retrieved knowledge and long-context reasoning. Too much external knowledge could degrade performance, yet I did not find any analysis or discussion of this trade-off in the paper.

• The experiments rely solely on GPT-4o-mini, which introduces uncertainty about whether the proposed mechanism only benefits weaker or outdated models. It is unclear whether the same strategy would remain effective with stronger modern models. This also calls the SOTA claim into question, since other methods combined with more capable models might outperform the proposed approach under equivalent settings.

• The method can be simplified to one sentence: when encountering difficulties, the system retrieves potentially relevant ideas from existing literature and recombines them into algorithmic solutions. This approach is essentially scientific knowledge reuse rather than the discovery of new algorithmic principles or heuristic paradigms. In essence, it guides the system toward an A + B = C style of engineering assembly, which is common in AI for Science Discovery research.

**Questions:**

Please refer to Weaknesses.

---

### Official Review · Reviewer_RfmV · 2025-10-26

**Soundness:** 3
**Presentation:** 2
**Contribution:** 2
**Rating:** 4
**Confidence:** 3

**Summary:**

This paper introduces Knowledge-Augmented Evolutionary Algorithm Design (KA-EAD), a novel framework for automatically discovering high-performance heuristics for NP-hard optimization problems. The idea is to make use of a knowledge base, and use a co-evolution outer loop to orchestract language model callings, eventually select phenotypes based on some problem dependent fitness function (if I understand the paper correctly).

**Strengths:**

Good illustration. The intricate design is also very thoughtful. Good amount of ablations in Table 1 and 2.

**Weaknesses:**

The idea isn't the most novel. The particular design is intricate and may be new to some degree, but such kind of LLM RAG enhancement has been a fairly crowded space where I already lost count of proper references.


A writing suggestion for the author is that there are too many big words/phrases that bear no useful information in scientific arguments. I don't know how much LLM helped in the writing process, but it's not writing a novel or business proposal. Using concise texts without too many superficial words can help save us readers' time (some examples: "The pivotal innovation ...", "synergistically...", etc.). It is also a good practice to minimize adverbs unless you truly want to assign a scientific attribute (e.g. "dramatically enhances" -> "enhances", since many readers prefer to be unbiased).

**Questions:**

1. Could you provide more detailed cost analysis for the entire pipeline in different situation? This has practical significance and would be very helpful context for people.
2. Biasing a search using a knowledge base has pros and cons. Have you explored whether it hurts diversity?
3. Have you ablated your own design components? Such as the choice of knowledge base, the need of reflection (vs rewrite), etc.?

---

### Official Review · Reviewer_qSZg · 2025-11-03

**Soundness:** 2
**Presentation:** 3
**Contribution:** 2
**Rating:** 4
**Confidence:** 5

**Summary:**

The paper proposes KA-EAD (Knowledge-Augmented Evolutionary Algorithm Design), a framework that enhances LLM-driven heuristic discovery for NP-hard problems by dynamically retrieving and integrating relevant scientific literature during evolution.

**Strengths:**

Originality & Significance:
The paper presents an innovative Knowledge-Augmented Evolutionary Algorithm Design (KA-EAD) framework that dynamically fuses LLM reasoning with scientific literature retrieval, marking a conceptual step forward in automated algorithm design and knowledge-driven AI research.

Technical Quality:
The methodology is rigorous and experiments are comprehensive across diverse optimization tasks, consistently outperforming strong baselines. The ablation and reproducibility details strengthen the paper’s credibility.

Clarity & Overall Evaluation:
The presentation is clear and well-organized, with logical structure and informative visuals. Minor stylistic tightening could improve flow.

**Weaknesses:**

While the proposed KA-EAD framework demonstrates strong performance, the paper does not sufficiently quantify its computational cost. Given that knowledge retrieval, reflection, and multi-round LLM calls can be resource-intensive, the paper would benefit from detailed comparisons of time, memory, and query cost against baseline methods (e.g., ReEvo, EoH). Including such analysis would clarify the trade-offs between performance gains and efficiency.

Although the framework leverages retrieved literature, the paper provides little analysis of what types of knowledge most effectively guide algorithm evolution or when retrieval fails. A qualitative study or case visualization showing examples of successful and unsuccessful retrievals would improve interpretability and help refine the reflection–retrieval mechanism for future iterations.

**Questions:**

Could the authors clarify how the reflection–retrieval–generation loop in KA-EAD determines when and what to retrieve from the literature during the evolutionary process? Specifically, is there a quantitative or heuristic criterion (e.g., stagnation in performance, uncertainty estimation, or semantic novelty detection) that triggers retrieval, or is it manually scheduled?

---

### Official Review · Reviewer_XTeu · 2025-11-03

**Soundness:** 3
**Presentation:** 2
**Contribution:** 2
**Rating:** 4
**Confidence:** 4

**Summary:**

This paper introduces KA-EAD, an evolutionary algorithm design (EAD) approach that augments conventional EAD techniques with a scientific knowledge retrieval module. This component enables the algorithm to draw on relevant external information beyond the limits of an LLM’s pre-training, guiding variation in candidate solutions. The main results are presented on the traveling salesman problem using classical and NCO solvers, both with and without LLM-based solutions (including the one proposed). Additional results on 15 NP-hard problems compared the proposed method to other LLM-based solutions. Ablation studies have been provided to analyze the contribution of different components.

**Strengths:**

The paper addresses the practical problem of ungrounded generations in LLM-based optimization, and the high-level solution proposed of retrieving chunks of prior literature is intuitive. A thorough description of the approach and implementation is provided, and useful ablation studies have been conducted to understand the benefit of the methodology.

**Weaknesses:**

1. Looking at the ablations in Figures 3 and (particularly) 5, it seems that most of the gains in performance are achieved due to the addition of reflections in the EA loop. Does this not counter the main message of the paper that internal knowledge of the LLM is insufficient? Or perhaps the chosen domains for evaluation are such that the retrieved knowledge is likely already baked into the pre-training of a modern LLM (e.g., o4-mini)? In the evaluated domains, it would be useful to understand (a) whether any of the retrievals actually make their way into the proposed solutions, and (b) whether the knowledge corpus contains useful articles *after* the cut-off date of the evaluated model that indeed help push performance beyond what internal knowledge could allow. To summarize, while it makes intuitive sense to add a retrieval step, I am concerned that the evaluations don't sufficiently demonstrate that benefit.
2. Results in Table 3 do not show the baseline performance of non-LLM solutions as in the main set of experiments from Tables 1 and 2. This would be useful in assessing (similar to Tables 1 and 2) the extent of improvement one is seeing in these various optimization problems. Additionally, while averages are reported in the tables, standard deviations are missing.
3. A small note on the writing -- it would be better for scientific exposition to cut several instances of laudatory adjectives that, in my opinion, are doing more harm than good, e.g., "pivotal", "innovative", "rigorous", "unprecedented capability". This is particularly noticeable in the Introduction.

**Questions:**

1. L193: is simple cosine similarity actually good here? Seems like we'd need more nuanced retrieval strategies to achieve what's shown in Figure 2 beyond using an off-the-shelf embedding model. Related (a): in L269, is $E_{chunk}$ also implemented as all-MiniLM-L6-v2? Related (b): how is \theta tuned?
2. What documents are used for $C_{lit}$?
3. L200: are chunks non-overlapping? If so, how is it ensured that chunk boundaries are sensible and don't split content that should be in the same chunk into different chunks?
4. In your experiments, how often is it the case that the final, improved solution was copied from the retrieval corpus instead of being "discovered" as a new generation by the LLM? Is the quantifiable with a small study?

---

### Official Review · Reviewer_CWJo · 2025-11-12

**Soundness:** 3
**Presentation:** 2
**Contribution:** 2
**Rating:** 4
**Confidence:** 4

**Summary:**

The paper demonstrates a combination of LLM‑driven evolutionary search with dynamic embedding-based retrieval from a curated scientific‑literature index. This allows the system to guide the LLM to better solutions to optimization problems like travelling salesman.  The authors show improvements over baselines that rely solely on LLM internal knowledge.

**Strengths:**

Good presentation and an interesting idea which has some novelty in applying LLMs and evolutionary algorithms to grounded algorithm discovery. The result is promising, but needs wider ablations to refine the core algorithm in order for it to be a significant advance.

**Weaknesses:**

- Only uses gpt-4o-mini, which means that the experimental results are weakly supported. What if other models, especially reasoning models which often behave differently?
- Constraining chunk length to 512 tokens seems like it could significantly limit the scope of possible insights the algorithm comes up with. Can you show ablations varying this?
- The ablations are generally low resolution and only seem to test 2 variants of the main algorithm (Figure 3). It's difficult from these to get a sense for how much each component is actually doing. I suspect the algorithm could be drastically simplified.

**Questions:**

- Why use an abstract embedding model rather than allowing an LLM to read all the abstracts and decide whether each deserves to be in the set using some metric?

---

### Meta-Review · Area_Chair_cMjv · 2026-01-04

**Summary:**

The reviewers agree that combining Evolutionary Algorithm Design (EAD) with knowledge retrieval is a reasonable and promising approach. Most of the concerns relate to the comparison with the existing literature, the idea's incrementality, the use of non-state-of-the-art models, and, finally, the overclaiming language in the paper. Although some of the reviewers do not provide references to the existing literature, the authors opted not to address the concerns raised and did not submit a rebuttal.

**Reviewer Concerns:**

None of the concerns are addressed because the rebuttal hasn't been submitted.

**Reviewer Scores:**

Reviewers would not change their scores due to the absence of the rebuttal.

---

### Decision · Program_Chairs · 2026-01-26

Reject